# Varietal Authenticity Assessment of QTMJ Tea Using Non-Targeted Metabolomics and Multi-Elemental Analysis with Chemometrics

**DOI:** 10.3390/foods12224114

**Published:** 2023-11-13

**Authors:** Huahong Liu, Yuxin Wu, Ziwei Zhao, Zhi Liu, Renjun Liu, Yuelan Pang, Chun Yang, Yun Zhang, Jinfang Nie

**Affiliations:** 1College of Chemistry and Bioengineering, Guilin University of Technology, Guilin 541004, China; 2College of Agriculture and Biotechnology, Hunan University of Humanities, Science and Technology, Loudi 417000, China; 3Guangxi Research Institute of Tea Science, Guilin 541004, China

**Keywords:** QTMJ tea, non-targeted metabolomics, multi-elemental analysis, chemometrics, varietal authenticity assessment

## Abstract

In this paper, a combination of non-targeted metabolomics and multi-element analysis was used to investigate the impact of five different cultivars on the sensory quality of QTMJ tea and identify candidate markers for varietal authenticity assessment. With chemometric analysis, a total of 54 differential metabolites were screened, with the abundances significantly varied in the tea cultivars. By contrast, the QTMJ tea from the Yaoshan Xiulv (XL) monovariety presents a much better sensory quality as result of the relatively more abundant anthocyanin glycosides and the lower levels of 2′-o-methyladenosine, denudatine, kynurenic acid and L-pipecolic acid. In addition, multi-elemental analysis found 14 significantly differential elements among the cultivars (VIP > 1 and *p* < 0.05). The differences and correlations of metabolites and elemental signatures of QTMJ tea between five cultivars were discussed using a Pearson correlation analysis. Element characteristics can be used as the best discriminant index for different cultivars of QTMJT, with a predictive accuracy of 100%.

## 1. Introduction

Tea originated in China and has traditionally been subdivided into green tea, yellow tea, white tea, oolong tea, black tea and dark tea. Green tea can maintain the natural substance in fresh leaves to a great extent and has become the second most popular and widely consumed drink in the world outside of water because of its attractive flavor quality and health benefits [1]. There are countless types of green tea in China from different geographical regions, cultivars and with various manufacturing processes. Qintang Maojian (QTMJ) tea is one of the top ten most popular Chinese green teas, which are produced in Qintang, Guangxi Zhuang Autonomous Region, China. QTMJ tea can date back over a thousand years. Tea plants for producing QTMJ tea are typically grown on the Tianping mountain, Songbai mountain and Zhuangmao mountain located in the Qintang district, where the mountains are covered by clouds and mist all throughout the year and the humidity in air is very high. The special climatic condition and soil nutrients which play important roles in the accumulation of bioactive compounds in tea give QTMJ tea a unique and exclusive flavor. QTMJ tea had been approved to be under the protection of geographical indications of agricultural products in China in February 2015 and has gained increasing attention.

The quality and chemical composition of tea is not only determined by the geo-graphical origins and the soil types, but is also closely associated with the processing techniques and the tea cultivars [2]. To date, several tea varieties are being cultivated in the region of Qintang, consisting of Longjing and Wuniu Zao introduced from the Zhejiang Province, Fuding Dabai and Fuyun 6 introduced from Fujian Province as well as Yaoshan Xiulv native to Guangxi. Traditionally, the fresh young buds and leaves from monovarieties are made into QTMJ tea through a series of processes including drying, blanching, kneading and aroma enhancing. In recent years, the tea market has been expanding rapidly owing to increased demand, bringing higher revenues and profits for tea growers and the industry. Meanwhile, the quality and price of tea are usually judged by the sensory assessment of professional tea tasters, which is highly subjective and lacks unified objective data support [3]. The phenomenon of fraud in the tea trade has become more and more serious due to the huge profits and subjective evaluations [4], QTMJ tea is no exception. For instance, deliberate mislabeling of tea cultivars or counterfeiting with inferior tea cultivars often appear in QTMJ tea and damage consumer trust. However, there is currently no report on the impact of different tea cultivars on the metabolomics and sensory quality of QTMJ tea, and the identification and assessment of tea cultivars. Therefore, in order to ensure traceability and authentication of premium QTMJ tea products, it is of urgent need to comprehensively evaluate the contribution of different tea cultivars to the quality and flavor of QTMJ tea and establish specific chemical markers for tea varietal identification.

Metabolic fingerprint analysis is a scientific method to explore the relationship between tea chemical composition and tea cultivars, which can generate large amounts of metabolic information and provide intensive insight into the intrinsic nature of macroscopic differences in different types of tea from the perspective of modern analytical chemistry [5]. Spectral detection technologies with the advantages of fast detection speed, low cost and non-destructiveness have been used for metabolic fingerprint analysis, such as fourier transform infrared spectroscopy (FT-IR), near-infrared spectroscopy (NIR), fluorescence spectroscopy, laser-induced breakdown spectroscopy (LIBS), terahertz time domain spectroscopy (THz-TDS), hyperspectral imaging, nuclear magnetic resonance spectroscopy (NMR) and Raman spectroscopy [6,7,8,9,10,11,12]. In comparison with spectral fingerprint technology, chromatographic fingerprinting, especially followed by MS analysis, since the detector has high sensitivity, specificity, reproducibility and is more suitable for qualitative analysis, shows that LC-MS-based non-targeted metabolomics as powerful analytical tools have been more and more accepted to characterize the chemical composition and quality assessment of tea [13,14]. Recently, Chen et al. [15] utilized the UPLC-QTOF/MS-based metabolomics approach coupled with multivariate statistical analysis to unveil the fundamental varietal differences of a broad range of metabolites among 14 major Wuyi Rock tea cultivars. A total of 49 primary metabolites were found to have clear variations between tea cultivars, of which catechins, kaempferol and quercetin derivatives were key metabolites for cultivar discrimination. Wang et al. [16] employed a UPLC-QTOF/MS-based widely targeted metabolomics approach to identify 54 candidate markers, which shows significant differences in expression among four different cultivars of Xinyang Maojian green tea. More recently, Zhao et al. [14] firstly attempted to identify markers for tea varietal authenticity assessment via the non-targeted UPLC-Q-Exactive Orbitrap-MS method combined with chemometrics. The results demonstrated that the seven highly similar oolong tea cultivars can be differentiated in terms of ten marker compounds with predictive accuracy equal to 89.8%. These studies based on untargeted metabolomics successfully revealed the composition differences of various tea cultivars and identified candidate markers for tea plant fingerprinting and cultivar identification according to the differences. Nevertheless, the metabolites can transform in a variety of ways during processing, transporting and storage and become highly similar among cultivars [17]. In fact, improving the effectiveness and accuracy of varietal authenticity assessment of various tea cultivars is still a challenging issue and needs to be further addressed. As is widely known, mineral elements as an indispensable part of the internal resources of plant life activities play a potential role in plant health and indirectly influence the accumulation of metabolites [18]. The multielement characteristics reflect the difference between the migration and transformation of mineral nutrients in the soil–plant system, closely related to the variety, climate, the soil type and rhizosphere environment [19]. Previous research has confirmed that the combination UPLC-QTOF/MS-based untargeted metabolomics with multi-element analysis provided a valid alternative to differentiate Chrysanthemum morifolium Ramat cv. “Hangbaiju” from different geographical origins [20] and to discriminate two varieties of eggplants [21]. At present, although mineral elements are important nutrients in tea [22], there are few studies on the varietal authenticity identification of different kinds of tea by using non-targeted UPLC-QTOF/MS metabolomics integrated with multi-element analysis, and the interaction relationship between mineral element composition and metabolites in green tea, especially QTMJ tea, remains largely unknown.

This study aims to (1) compare the variances of QTMJ tea metabolic profiles and element characteristics between cultivars and evaluate their quality; (2) screen out the significant variables based on statistical analysis and identify potential marker compounds for the cultivar authenticity of QTMJ tea with the aid of chemometrics; (3) explore the correlation between differential metabolites and elements based on the Pearson correlation coefficient. This study will open a new route for insights into the varietal diversity of QTMJ tea, providing effective help for the authenticity identification of tea cultivars and facilitating the sustainable development of the tea industry.

## 2. Materials and Methods

### 2.1. Chemical and Reagents

HPLC-grade acetonitrile was purchased from Shanghai Ampere Scientific Instruments Co., Ltd. (Shanghai, China), LC-MS-grade methanol was bought from Honeywell China Co., Ltd. (Shanghai, China) Hydrogen peroxide (H_2_O_2_, 30%) and nitric acid (HNO_3_, 65~68%) were purchased from Xilong Chemical Co., Ltd. (Guangzhou, China). Multi-elemental calibration standard #1 (As, Ba, Be, Bi, Cd, Co, Cu, Ga, Li, Mn, Ni, Pb, Sb, Sn, Sr, Tl, V, Zn, serial number GSB 04-1767-2004), standard #2 (Zr, Hf, W, Mo, Ta, serial number GSB 04-1768-2004), standard #3 (La, Ce, Pr, Nd, Sm, Eu, Gd, Tb, Dy, Ho, Er, Tm, Yb, Lu, serial number GSB 04-1789-2004), standard #4 (Th, serial number GBW (E) 080174) and standard #5 (U, serial number GBW (E) 080173) with the concentration of individual element 100 mg L^−1^ were purchased from the National Nonferrous Metals and Electronic Materials Analysis and Testing Center (Beijing, China) and the Beijing Research Institute of Chemical Engineering Metallurgy (Beijing, China), respectively.

### 2.2. Sample Collection and Pretreatment

In order to reduce the impacts of the variances of farming patterns and soil factors on the metabolites and elemental composition, all 30 first-grade samples of Qingtang Maojian (QTMJ) tea of five cultivars were produced by Fuwei Tea Co., Ltd. through the same manufacturing process in the spring of 2022 at Qintang district, China. The cultivars of Fuding Dabai (FD), Wuniu Zao (WNZ) and Longjing group species (LJ) are grown on Songbai Mountain (800 m, 109.43 E, 23.07 N), Fuyun 6 (FY) and Yaoshan Xiulv (XL) are planted on the Zhuangmao Mountain (600 m, 109.28 E, 23.13 N), all of which are located in the geographical indication protected area of QTMJ tea with the agricultural environment nearly consistent. The samples were ground to powder using a tissue grinder in an ice-bath environment and passed through a 100-mesh sieve for future analysis.

### 2.3. LC-MS Based Metabolomic Analysis

The tea powder (0.0200 g) was weighted accurately into a 10 mL centrifuge tube and mixed with 1.0 mL of 70% methanol aqueous solution (*v*/*v*). Then, the mixture was vortexed for 2 min and sonicated for 15 min at room temperature. The extract was centrifuged at 6000 rpm for 10 min and the analysis was performed in duplicate. The supernatant was filtered through 0.22 μm micron filter and immediately stored at −20 °C prior to HPLC-QTOF/MS analysis. In addition, the quality control (QC) samples were prepared by mixing equal quantities of all samples to validate the metabolomic methodology.

The metabolomic analysis of tea samples was performed by using Shimadzu LC-20A (Kyoto, Japan) equipped with a Sciex TripleTOF 5600+ in information-dependent acquisition (IDA) mode. A Waters XSelect HSS T3 column (3.5 μm, 2.1 mm × 150 mm) was used for chromatographic separations. An amount of 2 μL of each sample was injected onto the column. The mass spectrometer was operated in both positive and negative ionization modes. In the positive ion mode, 0.1% (*v*/*v*) formic acid was used as mobile phase A and pure acetonitrile were used as mobile phase B. Water with 5 mM ammonium acetate and pure acetonitrile were applied as mobile A and B in the negative ion mode. The gradient elution for the system is 0–3 min, 1% B; 3.01–24 min, 1–100% B; 24.01–32 min, 100% B; 32.01–37 min, 1% B. The mass spectrometer was scanned in the range of 50 to 1000 *m*/*z*. The source voltage and the collision energy were set to 5500 V and 30 V for the positive ion mode and 4500 V and −30 V for the negative ion mode, respectively.

The raw data files generated by HPLC-QTOF/MS were processed by using MS-DIAL software (version 4.36) to perform noising filtering, peak identification, over-lapped peak analysis, peak alignment and peak filling. The obtained MS peaks were identified by matching the retention time, mass accuracy, peak area and MS/MS fragmentation against online databases (e.g., MassBank, LipidBlast and MetaboBase) with an identification score cutoff of 80% and accurate mass tolerance of 0.05 Da for MS1 and 0.1 Da for MS2, respectively. The dataset containing the metabolites identified in both ionization modes was subsequently used for further statistical analysis.

### 2.4. Multielement Analysis

Two hundred milligrams of tea powder was accurately weighed into an acid-washed 50 mL Teflon digestion tube vessel. A mixed acid solution of 2 mL nitric acid (HNO_3_, 65~68%) and 1 mL hydrogen peroxide (H_2_O_2_, 30%) was added to the vessel and left to react for 20 min, the vessel was then placed in an electrothermal oven at 160 °C for 24 h digestion. After cooling, the digested sample solution was diluted to 10 mL with 2% HNO_3_. All samples were digested in triplicate. The elements were analyzed using a quadrupole-based inductively coupled plasma mass spectrometer (ICP-MS, model 7500cx, Agilent, Santa Clara, CA, USA) equipped with a collision/reaction cell (CRC). The optimized operating conditions for analysis are described in Appendix A.

### 2.5. Statistical Analysis

The significant difference between the metabolites and mineral elements among samples from different tea cultivars were determined by one-way analysis of variance (ANOVA) in SPSS statistics software (version 26.0, IBM, New York, NY, USA) with significance criteria set as *p*-value < 0.05. For quantitative analysis, all data were presented as mean ± SD, and the results were compared between the different groups.

Data were evaluated using hierarchical clustering analysis (HCA) and partial least squares discriminant analysis (PLS-DA) in SIMCA-P (version 14.1, Umetrics, Malmo, Sweden) as well as linear discriminant analysis (LDA) in SPSS statistics software. Venne diagram and heatmap visualization was performed using the OmicStudio tools at https://www.omicstudio.cn/tool (accessed on 16 September 2023). Pearson correlation analysis to explore the relationship between differential metabolites and mineral elements were carried out on the same Web site.

## 3. Results and Discussion

### 3.1. Difference of Metabolomic Fingerprints among QTMJ Tea Cultivars

Total Ion Flow Chromatograms (TICs) can reflect the overall information of the samples. The TICs of the QC samples in the positive and negative ion scanning modes are displayed in Appendix A and the peak shape and retention times are of a good coincidence, suggesting the robustness of the analytical procedure and the reliability of the obtained data. The differences between the chromatograms of the tea samples in both positive and negative ion modes can be found in Appendix A, denoting that the varietal differences in the metabolomic structures and composition are marked. After processing the HPLC-QTOF/MS-based metabolomic data by MS-DIAL according to the 80% threshold, a total of 442 metabolites were initially identified in both positive and negative ion modes and a five-set Venn diagram was constructed to visualize the number of the detected metabolites for various cultivars. As shown in Figure 1A, the number of identified metabolites is 190, 204, 187, 185 and 190 in FD, XL, FY, LJ and WNZ, respectively. And, there are 67 common metabolites and 172 specific metabolites, of which 30, 38, 45, 34 and 25 metabolites uniquely appear in FD, XL, FY, LJ and WNZ, respectively. This result, similar to the above-mentioned findings, further implies that the active compounds in QTMJ tea varied with the cultivars, leading to the existence of quality differences in the essence (Appendix A).

The principal component analysis (PCA) was used to characterize the metabolic profiles of the five cultivars. As illustrated in Appendix A, the samples from five cultivars are clearly clustered into two broad categories, FY is one group and FD, XL, LJ and WNZ gather into another group, meaning that there are documented significant differences in the metabolome in FY compared to other cultivars. The partial least squares discrimination analysis (PLS-DA) model (*R*^2^*X* = 0.874, *R*^2^*Y* = 0.792, *Q*^2^ = 0.571) was established to provide clearer differentiation between the samples. From Appendix A, it can be observed that an obvious separation trend exists in the FY, FD and XL, and yet the LJ and WNZ are still clustered into one group, declaring that the metabolite profile of LJ is similar to WNZ.

For further insight into the chemical similarity and differences between five QTMJ tea cultivars, the difference analysis based on the volcano map between any individual and the other cultivars was performed with the criteria of FC ≥ 2 and *p*-value ≤ 0.05, as shown in Figure 1B. By comparison, the number of differential metabolites is 42, 82, 52, 20 and 26 between FD and NFD, FY and NFY, LJ and NLJ, WNZ and NWNZ as well as XL and NXL, respectively. It is distinct that FY has the most differential metabolic compounds, providing direct evidence for its discrimination results yielded by PCA and PLS-DA. Moreover, according to the volcano maps between any pair of samples (Appendix A), it is further revealed that there exists no differential metabolic compound between LJ and WNZ. This result indicates that LJ samples might share a high similarity with WNZ, giving a relatively reasonable explanation for their unsatisfied separation in the PLS-DA analysis. In fact, some of the literature has confirmed that owing to their different manufacturing suitabilities, the LJ, FY and FD cultivars suitable for manufacturing mainly green tea, black tea and white tea, respectively, have distinctly different metabolite characteristics [23]. WNZ was born prematurely in Zhejiang Province, and is one of the earliest maturing green teas, whose sensory quality is very similar to LJ tea when they are processed according to the same manufacturing techniques, so WNZ has very similar chemical profiles to LJ, indistinguishable from each other.

### 3.2. Discriminating Cultivars of QTMJ Tea Based on Candidate Differential Metabolites

To better assess the varietal difference in the chemical properties of the five QTMJ tea cultivars, the differential metabolites were screened based on VIP > 1.5 and ANVOA (*p* < 0.01), and a total of 54 characteristic compounds with differences were found as the potential mark metabolites for distinguishing QTMJ tea of different cultivars, as shown in Table 1. Subsequently, to provide a more immediate view of these candidate markers, the 44 matched differential metabolites among them were subjected to a heatmap analysis merged with hierarchical cluster analysis (HCA).

Flavonoids and their glycosides are widely found as the main functional polyphenols in green tea, they have antioxidant and hypolipidemic effects and play an important role on the formation of flavor properties of tea [24]. As demonstrated in Table 1 and Figure 1C, the differential metabolites mainly involve flavonoids, terpenoids, alkaloids, organic acids and the most amount of which are flavonoid compounds. The flavanoles, (−)-epicatechin and (+)-catechin, have been found to result in a slightly astringent taste and a refreshing aftertaste in green tea infusions [25]. Both of them are present in much higher abundances in FY and FD and this accounts for their more refreshing sensory properties to some extent (Appendix A). In consideration of the better chemical stability of glycosides, flavonoids exist usually in the form of glycosides in tea [26]. Most of the differential flavonoid glycosides such as flavone glycosides, flavonole glycosides and anthocyanin glycosides exhibit relatively higher contents in XL, LJ and WNZ compared to the other two cultivars. Among them, XL samples have more abundant anthocyanin glycosides, leading to their tea infusion appearing more emerald-green and brighter. Previous studies indicated that the differences in the content of anthocyanin glycosides might be attributed to the variation in levels of the ANS (CSS0010687) gene expression determined by tea cultivars [27]. In general, flavone glycosides and flavonole glycosides are considered as key taste determinants, which have lower thresholds to generate velvety astringency tastes in tea infusions [28]. Nevertheless, it is worth noting that there are no significant differences between the abundances of vitexin-2-O-rhamnoside, kaempferol-3-rutinoside-4′-glucoside, apigenin-6,8-digalactoside and apigenin-8-C-glucoside-2′-rhamnoside in LJ and WNZ samples, which might cause their scores of taste evaluation to be close to each other and lead to several misjudged WNZ samples in LJ samples by HCA.

Furthermore, more than half of the differential terpenoids consisting of jasminoside, obacunone cafestol and dehydroandrographolide present significantly higher contents in FD and FY samples, however, the other terpenoid metabolites (ganoderic acid D2, ginsenoside Rg2 and ginsenoside Rg5) are relatively rich in XL, LJ and WNZ samples. Rinsenosides have strong anti-inflammatory effects [29] and both ginsenoside Rg2 and ginsenoside Rg5 are present in the highest abundances in XL samples. The triterpenoid metabolites have been reported to be dominantly synthesized via the mevalonic acid (MVA) pathway and MYB, MYC, bHLH, NAC, ERF and WRAKY were important transcription factors to regulate MVA pathway-related genes [30]. Hence, the differential expression of these transcription factors in different tea cultivars may have an important effect on the differential production of triterpenoid metabolites in QTMJ tea. Moreover, the corresponding abundances of a majority of alkaloids and organic acids are varied in different tea cultivars. The contents of 2′-o-methyladenosine, denudatine and kynurenic acid are higher in FY and FD samples, but the other alkaloid compounds are inversely higher in LJ, WNZ and XL samples, which might induce the bitterness of QTMJ tea. Additionally, L-pipecolic acid and kynurenic acid, which originated from lysine and tryptophan [31,32], are both identified in a relatively lower quantity in XL samples. It can therefore be inferred that the abundances of the bitter amino acids (lysine and tryptophan) are comparably lower in XL samples, possibly leading to the quality of XL with a fresher and mellow taste.

The peak areas of 54 candidate differential metabolites of QTMJ tea produced by five cultivars were employed to establish a supervised PLS-DA model for discriminative purposes and the results are illustrated in Figure 2A. The *R*^2^*Y* parameters describe the percentage of variation explained by the model and *Q*^2^ means the predictive ability of the model. Here, the parameters of *R*^2^*X*, *R*^2^*Y* and *Q*^2^ in the PLS-DA model are 0.954, 0.781 and 0.534, respectively, indicating that the PLS-DA model has a good classification capacity. Moreover, a cross-validation analysis with 200 permutation tests was performed to evaluate the reliability of the PLS-DA mode. The obtained intercepts of *R*^2^ and *Q*^2^ equal to 0.079 and −0.488 are lower than the original ones, respectively (Figure 2B), which denote no overfitting. However, although the clear separation of tea samples from FY, FD and XL groups can be found in the score plots, the PLS-DA model can only reach 80% accuracy for varietal discrimination, as the result of the high similarity between the WNZ and LJ cultivars. Linear discrimination analysis (LDA) as another supervised modeling method was attempted to reduce dimensionality and further improve the discrimination accuracies of QTMJ tea cultivars. The 54 candidate differential metabolites were further used for the LDA modeling, four linear discriminant functions and four class functions were built by simultaneously minimizing within-group variance and maximizing between-class variance of variables between samples and the score-scattering plot of tea samples projected on the coordinate system of first two discriminant function is shown in Appendix A. FD, FY and XL cultivars are completely separated and the classification accuracy increases from 80% to 93.3%, whereas the WNZ samples are still misjudged into the LJ group, which recall the above heatmap result that there are little gaps between the abundances of some flavonoid and terpenoid compounds. Therefore, though the non-targeted metabolomic analysis provides an opportunity for intensive insight into the metabolic fingerprints of different tea cultivars, effective discriminant analysis should be further investigated with the combination of more powerful tools to achieve a more robust and universally applicable cultivar identification.

### 3.3. Discriminating Cultivars of QTMJ Tea Using Elemental Fingerprinting

Mineral elements as an indispensable part of the internal resources of plant life activities can also reflect differences in the species within a single field [33]. Notably, they may be more suitable for tea authentication since they are less affected by processing and storage time and they are more stable [34]. Thus, the element characteristics based on ICP-MS were carried out for better differentiation between different tea cultivars. A total of 39 mineral elements were detected in QTMJ tea and the results were summarized in Table 2. It can be seen that the concentrations of 39 determined mineral elements except Zr and Hf in QTMJ tea samples are significantly different (*p* < 0.05 or *p* < 0.01) between any two of the five cultivars. The most abundant mineral element found in the five cultivars is Mn (824.42 ± 39.9 mg kg^−1^), followed by Zn (65.66 ± 2.62 mg kg^−1^) and Cu (18.06 ± 1.01 mg kg^−1^), with Tm, Lu, Ho, Tb, Ta and Hf being the least abundant. The contents of rare earth elements including La, Ce, Pr, Nd, Sm, Eu, Gd, Tb, Dy, Ho, Er, Tm, Yb and Lu are less than 200 μg kg^−1^ and the heavy metals such as Cd, Pb, Sn, As, Sb, Bi and Co are lower than 300 μg kg^−1^, both of which are far below the Chinese national standard GB 2762-2005 and NY 659-2003 [35], respectively.

Based on the examined 39 mineral elements, both PLS-DA (Figure 2C) and LDA (Appendix A) can achieve the purpose of discrimination of different tea cultivars. As shown in Figure 2C, the tea samples are well separated into five groups according to the tea cultivars with a discriminant accuracy of 100%. The results of PLS-DA with *R*^2^*X*, *R*^2^*Y* and *Q*^2^ high to 0.999, 0.967 and 0.954 indicate the presence of obvious variance in the mineral elements of tea samples. The reliability of the PLS-DA model was further verified by performing a cross-validation with 200 permutation tests (Figure 2D). The results with the intercepts of *R*^2^ and *Q*^2^ as 0.092 and −0.518 indicate that the PLS-DA model is reliable. The establishment of the LDA model based on 39 mineral elements also achieved excellent separation among different cultivars (Appendix A) with the discriminant accuracy equal to 100%.

The differential mineral elements of QTMJ tea for different cultivars were further screened with the aid of the PLS-DA loading plot. According to the criterion of VIP > 1 and *p* < 0.05, a total of 14 differential elements were selected among different cultivars, namely Li, V, Co, Ni, As, Sr, Mo, Cd, Ce, Yb, W, Tl, Th and U. Most of them play a positive role in tea plant growth and resistance, as well as bone development and disease prevention for human beings [36,37]. Fortunately, the concentrations of the harmful differential elements in all tea samples such as U and Cd, which are a greater threat to human health, are below the limits of national standards. To provide a more visual representation, a series of box plots were drawn to describe the distribution of the differential mineral elements in different tea cultivars. As presented in Figure 3, the XL samples have a relatively high abundance in the elements of Cd, W, Th, Ti, U, Ce, V, Ni, As, Sr and Mo, especially the concentrations of W, Th, U, V, Ni, As and Mo, which are far larger than the other cultivars. In fact, the XL cultivar has been planted on the same Zhuangmao Mountain with the FY cultivar under similar soil conditions and the same fertilizer management, however, the significant differences in the contents of the differential elements, except for Sr, between them can be observed. Moreover, similar results can be generated in the FD, LJ and WNZ cultivars, all of which has been cultivated on Songbai Mountain. These phenomena suggest that the genetic characteristics of the varieties may exert an important influence on the enrichment ability of elements. In other words, it is reasonable to guarantee the cultivar authenticity of QTMJ tea based on the screened potential marker elements by using elemental fingerprinting.

### 3.4. Correlation Analysis of Differential Metabolites and Mineral Elements

Research has found that mineral elements as indispensable nutrients play an important role in plant health and indirectly influence the accumulation of metabolites [20,38,39]. For the aim of exploring the relationship between main differential metabolites and mineral elements, a Pearson correlation analysis was carried out in this study. As depicted in Figure 4, there is a significant negative correlation between flavonoid glycosides and Tl, Li, Ce and Cd elements, with the absolute values of most of the correlation coefficients being greater than 0.5. In particular, Cd as a heavy metal is reported to affect the growth of plants and its accumulation in plants will lead to the production of reactive oxygen species (ROS). To counteract Cd stress, a series of antioxidant defense systems consisting of enzymatic reactions and non-enzymatic reactions will be activated to regulate ROS to the normal physiological level [40]. Glutathione transferase acts as a cofactor for enzymatic antioxidants in ROS quenching [41], and also plays a vital role in the accumulation of flavonoid metabolites [42]. The production of ROS induced by Cd might be a possible factor for effecting the accumulation of flavonoid glycosides, consistent with a significant negative correlation between Cd and flavonoid metabolites. In addition, the Co and Mo elements have been found to act like micronutrients in mitigating the toxicity of heavy metals [43], so a significant positive correlation between Co and flavonoid metabolites can be discovered in Figure 4. Furthermore, an obvious negative correlation can be found between amino acid derivatives and the Mo, As, W, Ni, Th and U elements; on the other hand, diphenylamine, cocamidopropyl betaine, ginsenoside Rg2 and these six mineral elements exhibit a significant positive correlation. The above results suggest that the metabolites are highly related to the enriched mineral elements in QTMJ tea, both of them contribute to the physiological processes in tea together, the mechanism of which needs to be explored in more detail in the future.

## 4. Conclusions

In the present paper, the varietal authenticity assessment of QTMJ tea was investigated by using LC-MS-based non-targeted metabolomics combined with multi-elemental analysis. A total of 54 differential metabolites were screened among the five QTMJ tea cultivars, most of which are flavonoids. Herein, the XL, LJ and WNZ cultivars are more suitable for manufacturing green tea, have larger levels of flavone glycosides, flavonole glycosides and anthocyanin glycosides. In particular, the QTMJ tea from the XL monovariety presents a more emerald-green and brighter quality as a result of the more abundant anthocyanin glycosides. The PLS-DA and LDA results show the high similarity of the metabolomic fingerprints in the LJ and WNZ cultivars, while the discriminant results based on the 14 differential mineral elements clearly show the excellent separation among the five cultivars, with a predictive accuracy of 100%. The Pearson correlation analysis confirmed the significant correlation between differential metabolites and elements in QTMJ tea, however, the interaction mechanism between mineral element composition and metabolites in green tea need to be further explored. In summary, metabolomics and multi-elemental analysis can be successfully employed for unveiling the impact of different cultivars on the sensory quality and the varietal identification of QTMJ tea, which is critically important to ensure the traceability and authentication of premium tea products and in turn contributes to the on-farm conservation of tea genetic diversity.

## Figures and Tables

**Figure 1 foods-12-04114-f001:**
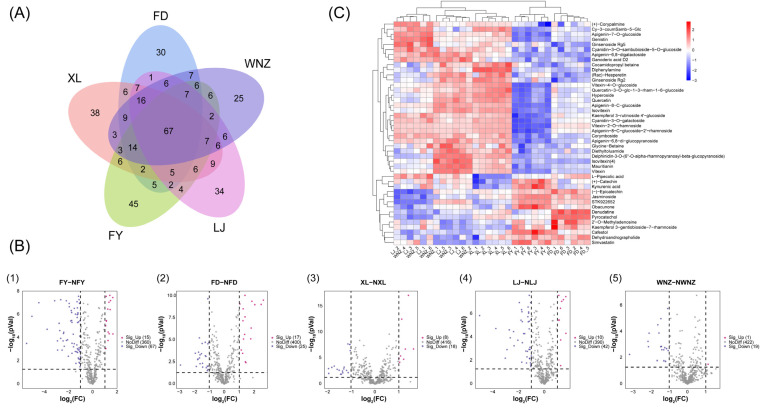
Metabolic profiles of QTMJ tea between five different cultivars. (**A**) Venn diagram; (**B**) volcanoes of metabolites between different cultivars (red and blue dots represent up-regulated and down-regulated differential metabolites, respectively; gray dots represent nondifferential metabolites); (**C**) heatmap and HCA analyses of metabolite contents in five QTMJ tea cultivars. (The redder the color, the higher the content; the more blue the color, the lower the content).

**Figure 2 foods-12-04114-f002:**
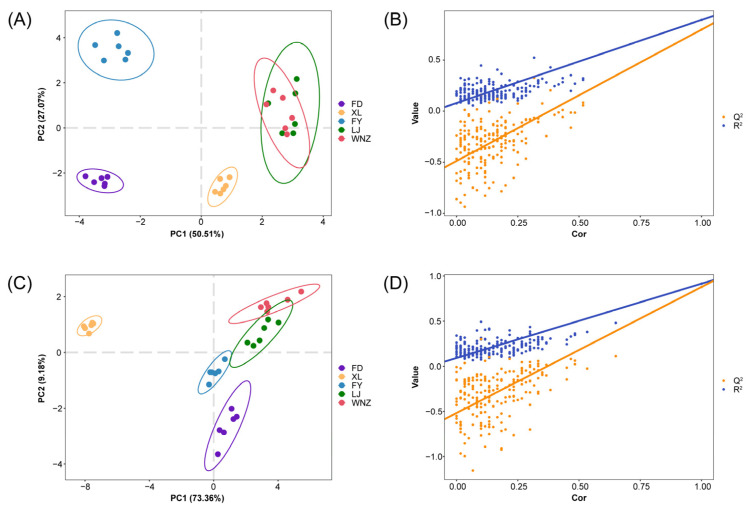
The discriminant results of the QTMJ tea cultivars. (**A**) PLS-DA score plot based on differential metabolites; (**B**) permutation test result of the score plot A; (**C**) PLS-DA score plot based on mineral elements; (**D**) permutation test result of the score plot B.

**Figure 3 foods-12-04114-f003:**
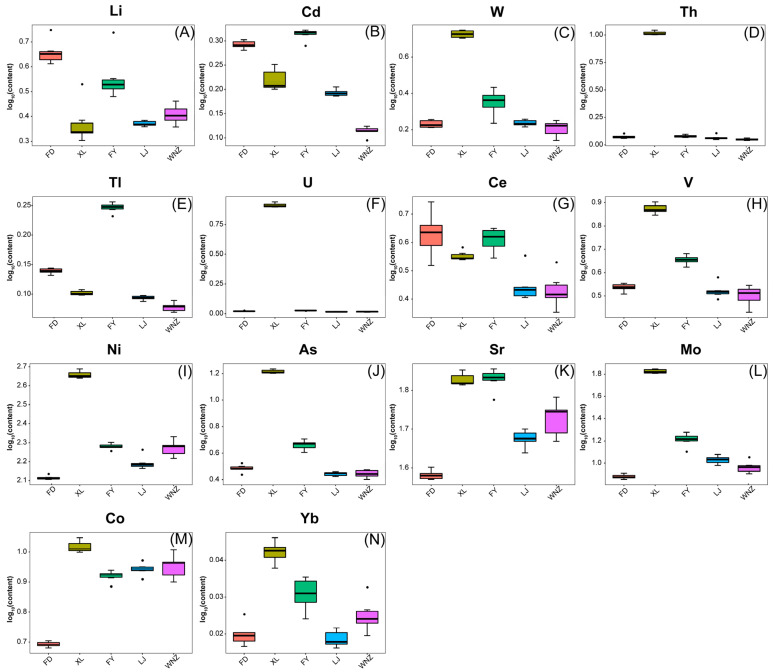
(**A**–**N**) Box plots of differential elements in different QTMJ tea cultivars.

**Figure 4 foods-12-04114-f004:**
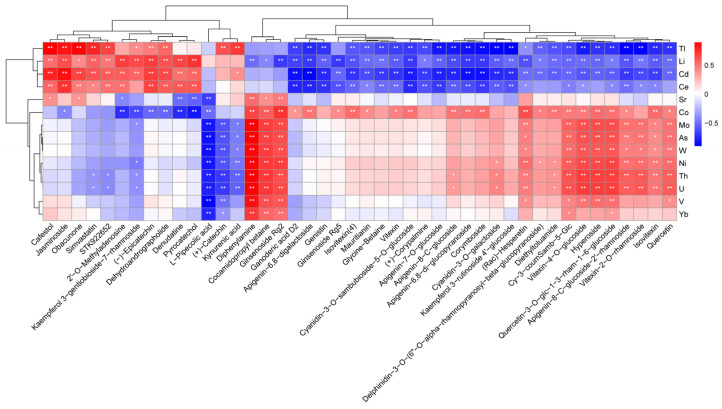
Pearson correlation analysis between 44 differential metabolites and 14 differential mineral elements. (blue indicates a negative correlation and red indicates a positive correlation, ** (*p* < 0.01) and * (*p* < 0.05) indicate the significance of correlation values).

**Table 1 foods-12-04114-t001:** Identified markers in QTMJ tea by metabolomics analysis based on LC-MS from negative ion and positive ion modes.

Metabolite Name	Classification	Adduct Type	Rt (Min)	M/Z	VIP	Significance
(+)-Catechin	Flavanole	[M+H]+	7.394	291.08478	1.60	**
(−)-Epicatechin	Flavanole	[M+H]+	8.956	291.0864	1.67	**
Isovitexin	Flavone	[M+H]+	8.637	433.10745	2.09	**
Isovitexin (4)	Flavone	[M+H]+	9.571	433.1131	2.14	**
Corymboside	Flavone	[M+H]+	8.881	565.1485	2.29	**
Vitexin	Flavone	[M+H]+	8.768	433.11374	2.05	**
Vitexin-4-O-glucoside	Flavone	[M+H]+	9.199	595.16187	2.05	**
Vitexin-2-O-rhamnoside	Flavone	[M+H]+	9.401	579.16632	1.58	**
Apigenin-6,8-digalactoside	Flavone	[M+H]+	7.865	595.16357	2.06	**
Apigenin-7-O-glucoside	Flavone	[M-H]−	10.399	431.09393	1.86	**
Apigenin-8-C-glucoside	Flavone	[M-H]−	9.262	431.09247	2.04	**
Apigenin-8-C-glucoside-2′-rhamnoside	Flavone	[M-H]−	9.212	577.15796	1.79	**
Apigenin-6,8-di-glucopyranoside	Flavone	[M-H]−	8.968	593.14679	2.24	**
STK922652	Flavone	[M+H]+	8.448	425.1402	1.87	**
Kaempferol 3-rutinoside 4′-glucoside	Flavonole	[M+H]+	9.138	741.21985	1.78	**
Kaempferol 3-gentiobioside-7-rhamnoside	Flavonole	[M-H]−	9.444	755.20581	2.01	**
Hyperoside	Flavonole	[M+H]+	9.199	465.09711	2.21	**
Quercetin	Flavonole	[M-H]−	11.887	301.03671	1.97	**
Quercetin-3-O-glc-1-3-rham-1-6-glucoside	Flavonole	[M+H]+	9.199	773.20691	2.23	**
Mauritianin	Flavonole	[M+Na]+	9.664	763.19934	1.81	**
(Rac)-Hesperetin	Flavanone	[M+H]+	10.928	303.08292	2.17	**
Genistin	Isoflavone	[M+H]+	10.419	433.11099	1.83	**
Cy-3-coumSamb-5-Glc	Anthocyanidin	[M]+	9.627	449.10898	2.21	**
Cyanidin-3-O-galactoside	Anthocyanidin	[M]+	9.442	889.25623	1.89	**
Cyanidin-3-O-sambubioside-5-O-glucoside	Anthocyanidin	[M-2H]−	7.177	741.18036	2.13	**
Delphinidin-3-O-(6″-O-alpha-rhamnopyranosyl-beta-glucopyranoside)	Anthocyanidin	[M]+	9.213	611.13898	1.72	**
(+)-Corypalmine	Alkaloid	[M+H]+	3.429	342.17606	1.64	**
2′-O-Methyladenosine	Alkaloid	[M+H]+	5.986	282.11865	2.13	**
Denudatine	Alkaloid	[M+H]+	19.91	344.25394	1.57	**
Kynurenic acid	Alkaloid	[M+H]+	7.699	190.04939	1.52	**
Cocamidopropyl Betaine	Alkaloid	[M+H]+	14.884	343.29813	2.13	**
Glycine Betaine	Alkaloid	[M+H]+	0.954	118.08591	1.62	**
Cafestol	Terpenoid	[M+H]+	19.939	317.20554	1.54	**
Dehydroandrographolide	Terpenoid	[M+H]+	16.4	333.20135	1.70	**
Jasminoside	Terpenoid	[M+H]+	12.07	303.0484	1.76	**
Ganoderic acid D2	Terpenoid	[M+Na]+	18.257	553.27069	1.53	**
Ginsenoside Rg2	Terpenoid	[M+H]+	24.279	785.49689	2.45	**
Ginsenoside Rg5	Terpenoid	[M-H]−	24.26	765.48004	1.94	**
Obacunone	Terpenoid	[M+Na]+	8.16	477.18555	1.80	**
L-Pipecolic acid	Organic acid	[M+H]+	1.076	130.08755	1.63	**
Diphenylamine	Other types	[M+H]+	17.737	170.09619	2.60	**
Pyrocatechol	Other types	[M+Na]+	15.327	553.29712	1.60	**
Simvastatin	Other types	[M+Na]+	18.648	441.25537	1.97	**
Diethyltoluamide	Other types	[M+H]+	14.488	192.13638	1.93	**
(carbon number 11)	Unknow	[M+H]+	2.211	229.15523	2.04	**
2-hydroxy-3-{[3,4,5-trihydroxy-6-(hydroxymethyl)oxan-2-yl]oxy}propyl(9Z,12Z,15Z)-octadeca-9,12,15-trienoate	Unknow	[M+H]+	20.007	515.3147	1.67	**
LPC 16:0	Unknow	[M+H]+	18.84	496.3378	1.59	**
LPC 18:3	Unknow	[M+H]+	17.294	518.32202	1.63	**
N-((S)-10-(((2S,3R)-1-(4-((4-chlorophenyl)sulfonyl)piperazin-1-yl)-3-methyl-1-oxopentan-2-yl)amino)-1,2,3-trimethoxy-9-oxo-5,6,7,9-tetrahydrobenzo[a]heptalen-7-yl)acetamide	Unknow	[M+Na]+	9.816	763.25299	1.99	**
NCGC00180744-03	Unknow	[M-H_2_O+H]+	9.975	331.15503	2.78	**
NCGC00380867-01	Unknow	[M+NH_4_]+	18.254	532.34662	1.54	**
NCGC00384602-01	Unknow	[M+Na]+	12.147	471.21854	2.09	**
NP-000062(6)	Unknow	[M-H]−	8.538	563.13953	1.93	**
(2S,3R,4S,5S,6R)-2-[(2E)-3,7-dimethylocta-2,6-dienoxy]-6-[[(2S,3R,4S,5S)-3,4,5-trihydroxyoxan-2-yl]oxymethyl]oxane-3,4,5-triol	Unknow	[M+FA-H]−	12.143	493.22742	2.31	**

Note: ** (*p* < 0.01) indicate a significant difference between five QTMJ tea cultivars.

**Table 2 foods-12-04114-t002:** The average values and ANOVA results of multi-element contents of QTMJ tea samples of five cultivars.

Elemental	FD	XL	FY	LJ	WNZ	VIP	Significance
Li [mg kg^−1^]	0.18 ± 0.03 ^a^	0.07 ± 0.03 ^c^	0.13 ± 0.04 ^b^	0.07 ± 0.00 ^c^	0.08 ± 0.01 ^c^	1.78	**
V [mg kg^−1^]	0.12 ± 0.01 ^c^	0.32 ± 0.02 ^a^	0.18 ± 0.01 ^b^	0.12 ± 0.01 ^c^	0.11 ± 0.02 ^c^	1.11	**
Mn [mg kg^−1^]	228.28 ± 6.97 ^d^	217.49 ± 8.56 ^d^	824.42 ± 39.9 ^a^	486.69 ± 21.94 ^c^	657.17 ± 65.73 ^b^	0.10	**
Co [mg kg^−1^]	0.20 ± 0.01 ^d^	0.47 ± 0.02 ^a^	0.37 ± 0.02 ^c^	0.39 ± 0.02 ^bc^	0.40 ± 0.04 ^b^	1.88	**
Ni [mg kg^−1^]	6.58 ± 0.16 ^d^	22.95 ± 1.02 ^a^	9.61 ± 0.35 ^b^	7.89 ± 0.68 ^c^	9.46 ± 0.91 ^b^	1.75	**
Cu [mg kg^−1^]	17.54 ± 0.49 ^a^	17.09 ± 0.64 ^ab^	18.06 ± 1.01 ^a^	16.92 ± 0.68 ^ab^	16.2 ± 1.51 ^b^	0.79	*
Zn [mg kg^−1^]	58.42 ± 1.63 ^b^	65.66 ± 2.62 ^a^	50.28 ± 3.12 ^c^	53.34 ± 2.14 ^c^	52.64 ± 5.07 ^c^	0.67	**
Ga [mg kg^−1^]	0.19 ± 0.00 ^e^	0.26 ± 0.01 ^c^	0.38 ± 0.01 ^a^	0.23 ± 0.02 ^d^	0.28 ± 0.02 ^b^	0.45	**
As [mg kg^−1^]	0.10 ± 0.01 ^c^	0.77 ± 0.03 ^a^	0.18 ± 0.02 ^b^	0.09 ± 0.00 ^c^	0.09 ± 0.01 ^c^	1.24	**
Zr [mg kg^−1^]	0.09 ± 0.10 ^a^	0.05 ± 0.00 ^a^	0.05 ± 0.00 ^a^	0.03 ± 0.00 ^a^	0.03 ± 0.01 ^a^	0.62	
Mo [mg kg^−1^]	0.33 ± 0.02 ^d^	3.29 ± 0.14 ^a^	0.77 ± 0.11 ^b^	0.49 ± 0.05 ^c^	0.41 ± 0.06 ^cd^	1.49	**
Cd [mg kg^−1^]	0.05 ± 0.00 ^a^	0.03 ± 0.00 ^b^	0.05 ± 0.00 ^a^	0.03 ± 0.00 ^b^	0.01 ± 0.00 ^c^	1.29	**
Sn [mg kg^−1^]	0.30 ± 0.01 ^ab^	0.33 ± 0.03 ^a^	0.31 ± 0.01 ^ab^	0.3 ± 0.03 ^b^	0.29 ± 0.04 ^b^	0.25	*
Sb [mg kg^−1^]	0.05 ± 0.01 ^ab^	0.03 ± 0.00 ^c^	0.06 ± 0.01 ^a^	0.03 ± 0.02 ^c^	0.04 ± 0.01 ^bc^	0.93	**
Ba [mg kg^−1^]	7.15 ± 0.24 ^d^	7.94 ± 0.33 ^c^	13.46 ± 0.75 ^a^	7.54 ± 0.51 ^cd^	9.16 ± 0.95 ^b^	0.13	**
La [mg kg^−1^]	0.10 ± 0.02 ^ab^	0.08 ± 0.03 ^abc^	0.10 ± 0.02 ^as^	0.07 ± 0.01 ^bc^	0.06 ± 0.03 ^c^	0.90	*
Ce [mg kg^−1^]	0.17 ± 0.04 ^a^	0.13 ± 0.01 ^b^	0.15 ± 0.02 ^ab^	0.09 ± 0.02 ^c^	0.09 ± 0.02 ^c^	1.10	**
Nd [mg kg^−1^]	0.04 ± 0.01 ^c^	0.04 ± 0.00 ^b^	0.05 ± 0.01 ^a^	0.03 ± 0.01 ^cd^	0.02 ± 0.00 ^d^	0.37	**
Gd [mg kg^−1^]	0.03 ± 0.01 ^bc^	0.02 ± 0.00 ^c^	0.04 ± 0.01 ^b^	0.02 ± 0.02 ^c^	0.06 ± 0.02 ^a^	0.15	**
W [mg kg^−1^]	0.04 ± 0.00 ^c^	0.22 ± 0.01 ^a^	0.06 ± 0.02 ^b^	0.04 ± 0.00 ^c^	0.03 ± 0.01 ^c^	1.23	**
Pb [mg kg^−1^]	0.29 ± 0.03 ^a^	0.26 ± 0.02 ^a^	0.26 ± 0.04 ^a^	0.16 ± 0.02 ^b^	0.13 ± 0.02 ^b^	0.82	**
Sr [μg kg^−1^]	1.86 ± 0.05 ^d^	3.31 ± 0.13 ^a^	3.32 ± 0.21 ^a^	2.32 ± 0.12 ^c^	2.63 ± 0.28 ^b^	1.28	**
Be [μg kg^−1^]	2.11 ± 0.56 ^cd^	3.02 ± 0.33 ^b^	6.80 ± 0.59 ^a^	1.85 ± 0.19 ^d^	2.48 ± 0.58 ^bc^	0.07	**
Pr [μg kg^−1^]	8.93 ± 1.96 ^bc^	10.03 ± 0.93 ^b^	13.37 ± 1.42 ^a^	7.21 ± 1.96 ^cd^	6.08 ± 0.71 ^d^	0.53	**
Sm [μg kg^−1^]	6.53 ± 1.13 ^b^	10.43 ± 0.41 ^a^	10.23 ± 1.06 ^a^	5.62 ± 1.12 ^bc^	5.23 ± 0.99 ^c^	0.26	**
Eu [μg kg^−1^]	3.43 ± 0.41 ^c^	4.00 ± 0.19 ^b^	6.41 ± 0.35 ^a^	3.51 ± 0.37 ^c^	4.04 ± 0.56 ^b^	0.12	**
Tb [μg kg^−1^]	0.72 ± 0.15 ^c^	1.64 ± 0.10 ^a^	1.12 ± 0.09 ^b^	0.62 ± 0.13 ^cd^	0.57 ± 0.10 ^d^	0.69	**
Dy [μg kg^−1^]	4.40 ± 0.59 ^c^	11.35 ± 0.65 ^a^	6.56 ± 0.63 ^b^	3.63 ± 0.50 ^cd^	3.57 ± 0.85 ^d^	0.86	**
Ho [μg kg^−1^]	0.68 ± 0.07 ^c^	1.92 ± 0.11 ^a^	1.12 ± 0.12 ^b^	0.92 ± 0.72 ^bc^	0.62 ± 0.12 ^c^	0.98	**
Er [μg kg^−1^]	2.10 ± 0.15 ^c^	5.98 ± 0.40 ^a^	3.29 ± 0.38 ^b^	1.78 ± 0.35 ^c^	1.91 ± 0.52 ^c^	0.96	**
Tm [μg kg^−1^]	0.22 ± 0.04 ^c^	0.69 ± 0.12 ^a^	0.34 ± 0.02 ^b^	0.16 ± 0.08 ^c^	0.21 ± 0.10 ^c^	0.73	**
Yb [μg kg^−1^]	2.34 ± 0.37 ^d^	5.10 ± 0.36 ^a^	3.68 ± 0.54 ^b^	2.19 ± 0.27 ^d^	2.96 ± 0.54 ^c^	1.25	**
Lu [μg kg^−1^]	0.19 ± 0.05 ^c^	0.58 ± 0.07 ^a^	0.33 ± 0.07 ^b^	0.16 ± 0.04 ^c^	0.19 ± 0.08 ^c^	0.90	**
Hf [μg kg^−1^]	2.05 ± 1.75 ^a^	1.49 ± 0.11 ^ab^	1.36 ± 0.17 ^ab^	1.01 ± 0.16 ^b^	0.97 ± 0.28 ^b^	0.61	
Ta [μg kg^−1^]	0.80 ± 0.24 ^a^	0.48 ± 0.05 ^bc^	0.45 ± 0.03 ^c^	0.60 ± 0.14 ^bc^	0.63 ± 0.09 ^b^	0.90	**
Tl [μg kg^−1^]	18.93 ± 0.71 ^b^	13.32 ± 0.57 ^c^	38.13 ± 1.68 ^a^	12.12 ± 0.53 ^d^	9.93 ± 1.01 ^e^	1.22	**
Bi [μg kg^−1^]	9.60 ± 0.63 ^c^	20.58 ± 0.97 ^a^	10.82 ± 0.77 ^b^	4.50 ± 0.37 ^e^	5.78 ± 0.81 ^d^	0.49	**
Th [μg kg^−1^]	9.30 ± 2.20 ^b^	469.52 ± 20.97 ^a^	9.88 ± 1.54 ^b^	8.18 ± 2.77 ^b^	6.00 ± 1.00 ^b^	1.33	**
U [μg kg^−1^]	2.57 ± 0.23 ^b^	357.96 ± 16.47 ^a^	3.23 ± 0.21 ^b^	1.96 ± 0.32 ^b^	1.92 ± 0.31 ^b^	1.39	**

Note: Data are shown as the mean ± standard deviation. Different lowercase superscripts represent the significant difference between any two cultivars at *p* < 0.05. ** (*p* < 0.01) and * (*p* < 0.05) indicate a significant difference between five QTMJ tea cultivars.

## Data Availability

The original contributions presented in the study are included in the article, further inquiries can be directed to the corresponding authors.

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
