# Peer review of "Varietal Authenticity Assessment of QTMJ Tea Using Non-Targeted Metabolomics and Multi-Elemental Analysis with Chemometrics"

_foods, 2023, doi:10.3390/foods12224114_

Round 1
Reviewer 1 Report
Comments and Suggestions for Authors
The publication titled: "Identification of Qintang Maojian green tea of different culti-..." is a very good paper. Apart from a few minor corrections, I think it is suitable. However, I recommend a few minor corrections and additions.
Below are my questions, comments, and suggestions:
1. The title could probably be slightly shortened.
2. Is the first sentence in the abstract necessary? This would probably be more useful for the authors as an introduction. However, the authors also start the introduction in a similar tone. So why duplicate content?
2a. "Therefore, this strategy is promising to be a feasible method for verifying the cultivars of QTMJ tea, 25 ensuring the development of the tea industry." I guess this sentence at the end of the abstract is also unnecessary.
3. Does "QTMJ tea" such a keyword make sense? Let the authors defend it somehow.
4. Couldn't Graphical Abstract (although very cool and creative!) be made with better resolution and care? I'm just asking.
5. To the introduction: a very important question - has no one really tried to examine green tea extracts using infrared spectroscopy - FTIR? That would be super interesting. Please add one or two sentences on this topic to the introduction, plus, of course, appropriate citations. As I see, there is quite a bit of it on the Internet. Additionally, this spectroscopic method complements mathematical chemometrics very well.
6. Fig 1 - what does Panel C give me in this drawing? You can't see anything there even after magnifying it. Maybe make one figer out of it? Some separate one? Panel B is also completely illegible. I understand that the authors later describe it in the text and I agree with that, everything is written correctly, but Figers himself is terribly illegible.
7. Fig 2 - captions on the axes are not visible - please correct it.
8. Fig 3 - doesn't the Y axis need to be signed here?
9. The last sentence in the abstract - it's weird. Too general and rather unnecessary. Or please write them in a more sensible way.
To sum up, it's a nice work, but you still need to work on it for a while.
Reviewer 2 Report
Comments and Suggestions for Authors
1. Introduction
The authors should introduce the available reported analytical method for tea authentication. There are many options to do authentication, especially for tea from HPLC to spectroscopic-based methods. Then, show us the strong reason why using metabolomics and multi-elemental.
Also briefly show us the reason why using this combination method.
2. Materials and Methods
Why the authors did not provide the adulterated tea samples for authentication purposes? For example, is it possible for the developed methods based on metabolomics and multi-elemental to authenticate the tea fraud due to adulteration?
3. Results and discussion
Fig. 1 B and C are difficult to read. Please provide more readable images.
In the classification result, the authors should provide clear information on how they prepare the samples for developing models and predictions.
The input variables for the LDA are not clear. Please provide the information on how the authors select the candidate for LDA input.
Round 2
Reviewer 2 Report
Comments and Suggestions for Authors
The revision is acceptable.